

# Segmentation of biventricle in cardiac cine MRI via nested capsule dense network

Jilong Zhang[1,*],Yajuan Zhang[1,*],Hongyang Zhang[1],Quan Zhang[2,3],Weihua Su[1],Shijie Guo[4] and Yuanquan Wang[1,4]

[1] School of Artificial Intelligence, Hebei University of Technology, Tianjin, China
[2] School of Information and Communication Engineering, North University of China, Taiyuan, China
[3] Shanxi Provincial Key Laboratory for Biomedical Imaging and Big Data, North University of China, Taiyuan, China
[4] Hebei Key Laboratory of Robot Perception and Human-Robot Interaction, HeBUT, Tianjin, China
[*] These authors contributed equally to this work.

## ABSTRACT

**Background**. Cardiac magnetic resonance image (MRI) has been widely used in diagnosis of cardiovascular diseases because of its noninvasive nature and high image quality. The evaluation standard of physiological indexes in cardiac diagnosis is essentially the accuracy of segmentation of left ventricle (LV) and right ventricle (RV) in cardiac MRI. The traditional symmetric single codec network structure such as U-Net tends to expand the number of channels to make up for lost information that results in the network looking cumbersome.

**Methods**. Instead of a single codec, we propose a multiple codecs structure based on the FC-DenseNet (FCD) model and capsule convolution-capsule deconvolution, named *Nested Capsule Dense Network (NCDN)*. NCDN uses multiple codecs to achieve multi-resolution, which makes it possible to save more spatial information and improve the robustness of the model.

**Results**. The proposed model is tested on three datasets that include the York University Cardiac MRI dataset, Automated Cardiac Diagnosis Challenge (ACDC-2017), and the local dataset. The results show that the proposed NCDN outperforms most methods. In particular, we achieved nearly the most advanced accuracy performance in the ACDC-2017 segmentation challenge. This means that our method is a reliable segmentation method, which is conducive to the application of deep learning-based segmentation methods in the field of medical image segmentation.

Corresponding authors
Quan Zhang, zhangib-met@nuc.edu.cn
Yuanquan Wang, wangyuanquan@scse.hebut.edu.cn

# INTRODUCTION

Heart disease causes one-third of all deaths worldwide. The statistics of the World Health Organization in 2016 proved that cardiovascular disease accounted for 31% of the world's total deaths (*Mozaffarian et al., 2015*). It is predicted that by the year 2030, a population of 23.3 million will be killed by cardiovascular diseases (CVDs) all over the world (*World Health Organization, 2010*; *Mathers & Loncar, 2006*). With the wide application of modern

medical technology, *i.e.,* magnetic resonance imaging (MRI), making a noninvasive qualitative and quantitative evaluation of cardiac anatomical structure and function has become increasingly convenient. At the same time, researchers have invested a lot of effort in the research of cardiovascular diseases to find out effective methods to reduce morbidity and mortality in recent years.

The regression method such as direct and simultaneous four-chamber volume estimation by the multioutput sparse latent regression (MSLR) (*Zhen et al., 2015*), use DAISY feature to train the regression model (*Du et al., 2018*) and Contour-Guided Regression Models (*Wang et al., 2019*), has been employed to predict the ventricular functional indices, while the most popular way to estimate the functional indices is based on segmentation, *i.e.,* segmentation of the ventricles first and then calculating the indices. The calculation of related indicators relies on the manual and accurate depiction of the endocardial and epicardial contours of the left ventricle (LV) and right ventricle (RV) (*Tran, 2016*). The continuous optimization of models and segmentation methods has made great contributions to the improvement of accuracy (*Wu, Wang & Jia, 2013*; *Zhang et al., 2020*). Manual rendering is a time-consuming and tedious task and is prone to high variability within and between observers (*Petitjean & Dacher, 2013*; *Miller et al., 2013*; *Tavakoli & Amini, 2013*; *Suinesiaputra et al., 2014*). Therefore, it would be very helpful to find a fast, high accuracy, reusable, automatic segmentation.

Before the rise of deep learning, some methods, such as threshold-based segmentation (*Sadeghian et al., 2009*; *Bazin & Pham, 2008*), edge detection-based segmentation (*Belaid & Mourou, 2009*; *Bellon & Silva, 2002*), and genetic algorithm-based segmentation (*Ramos & Muge, 2004*) cannot compare with deep learning-based segmentation methods in effect.

Although deep learning-based cardiac MRI segmentation has made great progress in the past decades, there are still many problems to be solved. So far, a model has not been found that is generally applicable to cardiac MRI segmentation tasks in various scenarios. The existing heart datasets have the problems of a small amount of data and insufficient data distribution so that the trained model does not respond well to the real-world situation, resulting in insufficient generalization ability. It has become the goal of many researchers to construct a model with fast learning speed and strong generalization ability on a limited data set.

In this article, we firstly propose a nested neural network architecture named Nested Capsule Dense Network (NCDN), which combines the FC-DenseNet model (*Jégou et al., 2016*) and capsule convolution-capsule deconvolution (*Lalonde & Bagci, 2018*). The Capsule Dense Block (CDB) is an important component module, which consists of a dense connection of multiple Capsule Convolution Units (CCU). Each CCU contains two capsule convolution layers and a capsule deconvolution layer. Because the introduction of the capsule model eliminates the traditional pooling layer for image size scaling and can retain more information for further semantic confirmation, the convolution capsule-deconvolution capsule is used to replace the convolution-deconvolution to implement CCU (*Sabour, Frosst & Hinton, 2017*). The correctness of this choice was also proved by subsequent experiments. The nested capsule dense architecture intuitively decomposes a

single codec structure into multiple sub-codec structures and uses the dense structure to better integrate feature information. Each step of feature extraction and reconstruction in CDB is accompanied by the abstraction and materialization of features by the network. Through multiple encoding and decoding, the image noise contained in the feature map is filtered out layer by layer, so that the network can learn more general features, such as contour features. On this basis, we can improve the generalization ability of the network. This nested network structure is our attempt to improve the FC-DenseNet model, which must be a trade-off between the accuracy of local marking and the determination of semantics. Furthermore, we replace the "concat" operation in the FC-DenseNet model with an "add" operation to reduce the parameters of the model. The verification effect on the test sets proves that our NCDN has better performance and stronger robustness than other models. There are four main contributions to our work, which can be summarized as follows:

(1) We firstly propose a nested capsule dense network called NCDN to decompose a single codec into multiple codecs. This structure allows more posture and other information to be retained and the noise in the sample is easier to filter in the early stage of feature learning.

(2) The Capsule Dense Block made of Capsule Convolution Units (CCU) designed by us eliminates the traditional pooling layer for image size scaling and can retain more information for further semantic confirmation.

(3) We design a new connection structure, which greatly reduces the model parameters.

(4) The proposed NCDN model is used to complete the segmentation task of cardiac MRI. The bi-ventricular segmentation and cardiac function diagnosis tasks in the ACDC 2017 dataset have shown good results.

## RELATED WORK

In the past few years, the segmentation of bi-ventricle MR images has received considerable attention. Many scholars have proposed various methods to obtain better accuracy and make the model have stronger generalization ability. The main model types can be summarized as Fully Convolutional Neural Networks (FCNs), Recurrent Neural Networks (RNNs), and Generative Adversarial Networks (GAN) (*Chen et al., 2019*).

By replacing the fully connected layer in the classification network with a convolutional layer, FCN (*Long, Shelhamer & Darrell, 2015*) predicts the category of each pixel in a dense prediction manner, which is successfully applied to the field of image segmentation and has become the mainstream method of current ventricular segmentation. *Tran (2016)* demonstrated the effectiveness of a fully convolutional neural network architecture for semantic segmentation of cardiac MRI and the utility of FCN to segment complex features of the left and right ventricles. Moreover, in order to reduce the class imbalance problem in ventricle segmentation and reduce the computational cost, *Abdeltawab et al. (2020)* used two FCNs (*Long, Shelhamer & Darrell, 2015*) to complete the selection of the region of the interest (ROI) and segmentation of instances. Different from ordinary FCN, *Ronneberger, Fischer & Brox (2015)* proposed a multi-scale U-shaped network based

on channel connections to refine the segmentation results. In order to avoid the loss of information caused by the maximum pooling layer in U-Net, *Shen et al. (2020)* use a purely dilated convolution operation to increase the receptive field to accelerate model convergence and improve model performance. *Sun et al. (2020)* believe that shape information is more meaningful than image texture information and thus add a secondary stream that processes shape features of the image in parallel with the U-Net to help cardiac ventricular segmentation. Taking advantage of the implicit deep supervision and feature reuse of the dense connection mechanism, *Jégou et al. (2016)* extended DenseNet (*Huang et al., 2016*) to FC-DenseNet for semantic segmentation problems. Similarly, *Penso et al. (2021)* proposed a dense block-based skip connection structure to reduce the semantic gap of skip connections in ventricular segmentation.

Unlike natural images, many medical images are 3D time series made up of 2D images, such as CT and MRI. 3D UNet (*Özgün et al., 2016*) and 3D VNet (*Milletari, Navab & Ahmadi, 2016*) extend the 2D segmentation network to the 3D segmentation network, making better use of the temporal and spatial information present in cardiac data to achieve accurate segmentation. However, recurrent neural networks, especially LSTM (*Hochreiter & Schmidhuber, 1996*) and GRU (*Cho et al., 2014*) have more advantages than convolutional neural networks for processing time-series tasks. One of the use cases is *Poudel, Lamata & Montana (2017)* combined RNN and 2D FCN to exploit the observed spatial dependencies in adjacent slices, improve the model's ability to identify the border regions of the heart, and solve the segmentation problem of multi-slice MRI images in a straightforward manner. GAN network is a competitively aware network structure, which is generally composed of generators and discriminators. In the process of model training, the generator generates images that attempt to deceive the discriminator, and the discriminator aims to identify real images in fake images. In the application of heart image segmentation, the role of the segmentation network is to generate segmentation results, and the discriminator is used to judge the difference between segmentation results and ground truth. *Qi et al. (2019)* adopted an adversarial training approach, where the generator and discriminator optimized the network by competing with each other, which alleviated the class imbalance of the heart, eliminated interference from other organs and tissues, and improved the segmentation accuracy of general "difficult" slices. In this way, a more accurate segmentation map will be generated.

*Sabour, Frosst & Hinton (2017)* proposed a capsule network (CapNet) with dynamic routing to use the reconstruction of output capsule instead of maximum pooling. The vector output of CapNet is better than the scalar output of Convolutional Neural Networks (CNN) to discover and save the position and posture information of objects in the image (such as spatial angle magnitude order, *etc.*). However, although CapNet has obtained good results in digital recognition and small image recognition, it has the problem of large parameters when performing large-scale image segmentation tasks or deep network construction, while images in the medical field are mostly large-size images. Therefore, the original CapNet is not suitable for image segmentation tasks in the medical field. *Lalonde & Bagci (2018)* modified the capsule network and applied it to the image segmentation task for the first time. They improved the dynamic routing algorithm to reduce the parameters.

The dynamic routing in the traditional capsule network is equivalent to the full connection mapping between capsules, which makes the number of parameters huge. The author uses window control and the same type of capsule sharing weight method to reduce the parameters. In addition to changing the dynamic routing algorithm to increase the size of the accepted input picture, a novel capsule convolution-capsule deconvolution network architecture called SegCaps is proposed to perform image segmentation tasks. Based on work (*Lalonde & Bagci, 2018*), *Cao et al. (2021)* proposed to extract low-level image features such as grayscale and texture of the left ventricle of the heart, as well as semantic features such as location and size for ventricular segmentation.

Inspired by the FC-DenseNet model (*Lalonde & Bagci, 2018*) and the SegCaps model (*Lalonde & Bagci, 2018*), our model was proposed for the semantic segmentation task of cardiac MRI.

## MATERIALS & METHODS

### Network structure

We will introduce our network in detail. As shown in Fig. 1, based on FC-DenseNet,

(1) We replace the dense block with Capsule Dense Block (CDB) proposed by us, this block will be explained in detail in 'Capsule Convolution Unit'.

(2) We design a new connection structure: Define $y_k$ as the $k$th layer. When $y_{k-1}$ is Transition Down (TD), $y_k$ is CDB, and $y_{k+1}$ is

$$y_{k+1} = y_{k-1} + y_k \tag{1}$$

where "+" means that the feature maps produced in layers $k-1$, $k$ are added in the last dimension. This means that the shape of the $k-1$th and $k$th layers must be consistent.

By fusing the output feature map of the CDB and the input feature map again, the relationship between layers can be closer, and the whole semantics cannot be directly connected in the process of capturing, to reduce the omission of image information. The other parts are similar to the idea of FC-DenseNet. Convolution is used to extract feature images, TD reduces the image size to increase the perception range, and Transition Up (TU) performs image reconstruction and precise positioning.

The detailed structure is shown in Table 1. We increase its channel number to 32 in the first convolution operation and do not change it in the subsequent process until it is finally transformed into the target number of channels through three convolutions. The purpose of our design is to avoid excessive parameters. The method we adopted is that the input and output of the CDB are the same in shape. If only "concat" is used instead of "add", the parameter increases from 5.5M to 330.8M, which is due to the dense feature of nested structure.

### Capsule convolution unit

The CCU is a constituent element of the CDB. By using the capsule convolution and capsule deconvolution structure proposed by *Lalonde & Bagci (2018)* to achieve the nested encoding-decoding structure in the CDB. The structure of a single CCU is as shown in Fig. 2 shown.

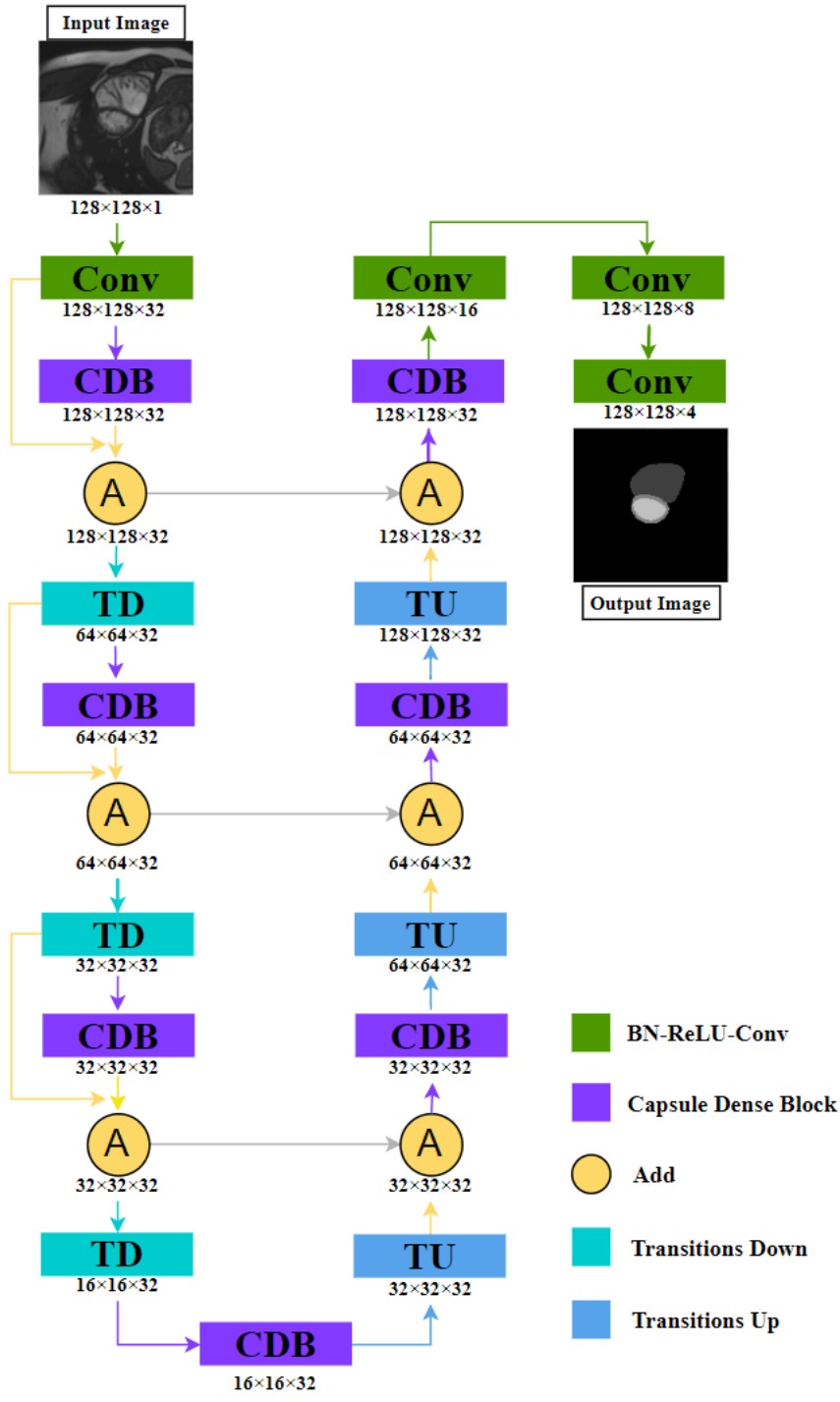

**Figure 1** The illustration shows the NCDN architecture used for ACDC segmentation tasks, the output image consists of four feature maps that represent background, RV, MYO, and LV, respectively. The data shown in the figure comes from the ACDC dataset.

**Table 1  The architecture of the Nested Capsule Dense Network. Conv layer in the table represents the BN-ReLU-Conv sequence.**

| Layers | NCDN | Ouput size |
|---|---|---|
| Input size | $128 \times 128 \times 1$ | – |
| Convolution | $3 \times 3$ Conv, stride 2 | $128 \times 128 \times 32$ |
| Capsule Dense Block | [BN-ReLU-CCU-Dropout] $\times 3$ | $128 \times 128 \times 32$ |
| Add | Convolution + Capsule Dense Block | $128 \times 128 \times 32$ |
| Transition Down | BN-ReLU-$2 \times 2$ max pool, stride 2 | $64 \times 64 \times 32$ |
| Capsule Dense Block | [BN-ReLU-CCU-Dropout] $\times 3$ | $64 \times 64 \times 32$ |
| Add | Convolution + Capsule Dense Block | $64 \times 64 \times 32$ |
| Transition Down | BN-ReLU-$2 \times 2$ max pool, stride 2 | $32 \times 32 \times 32$ |
| Capsule Dense Block | [BN-ReLU-CCU-Dropout] $\times 3$ | $32 \times 32 \times 32$ |
| Add | Convolution + Capsule Dense Block | $32 \times 32 \times 32$ |
| Transition Down | BN-ReLU-$2 \times 2$ max pool, stride 2 | $16 \times 16 \times 32$ |
| Capsule Dense Block | [BN-ReLU-CCU-Dropout] $\times 3$ | $16 \times 16 \times 32$ |
| Transition Up | $3 \times 3$ deconv, stride 2 | $32 \times 32 \times 32$ |
| Add | Deconvolution + Capsule Dense Block | $32 \times 32 \times 32$ |
| Capsule Dense Block | [BN-ReLU-CCU-Dropout] $\times 3$ | $32 \times 32 \times 32$ |
| Transition Up | $3 \times 3$ deconv, stride 2 | $64 \times 64 \times 32$ |
| Add | Deconvolution + Capsule Dense Block | $64 \times 64 \times 32$ |
| Capsule Dense Block | [BN-ReLU-CCU-Dropout] $\times 3$ | $64 \times 64 \times 32$ |
| Transition Up | $3 \times 3$ deconv, stride 2 | $128 \times 128 \times 32$ |
| Add | Deconvolution + Capsule Dense Block | $128 \times 128 \times 32$ |
| Capsule Dense Block | [BN-ReLU-CCU-Dropout] $\times 3$ | $128 \times 128 \times 32$ |
| Convolution | $3 \times 3$ Conv, stride 2 | $128 \times 128 \times 16$ |
| Convolution | $3 \times 3$ Conv, stride 2 | $128 \times 128 \times 8$ |
| Convolution | $3 \times 3$ Conv, stride 2 | $128 \times 128 \times 4$ |

The specific details are: let the input length, width, number of capsules, and number of channels be $K_h$, $K_w$, $K_{\text{cap}}$, $K_c$. The input [ $K_h$, $K_w$, $K_{\text{cap}}$, $K_c$] first passes through $3 \times 3$ capsule convolution with routing number 1, which becomes [ $K_h/2$, $K_w/2$, $K_{\text{cap}} \times 2$, $K_c$]. After passing $3 \times 3$ capsule convolution with routing number 3, it becomes [ $K_h/2$, $K_W/2$, $K_{\text{cap}}$, $K_c$]. Finally, the shape of the output is [ $K_h$, $K_w$, $K_{\text{cap}}$, $K_c$] after $3 \times 3$ capsule deconvolution with routing number 3.

## Capsule dense block

Capsule Dense Block (CDB) in Fig. 3 consists of a dense capsule connection layer at the front and a regression layer at the rear. The dense capsule connection layer is composed of three Capsule Convolution Units (CCU) densely connected. The regression layer is a convolutional layer and its purpose is to convolve the feature map ($F_{d,4}$) formed by the dense connection into the shape of the CDB input ($F_{d,1}$).

Dense connection: in CDB, dense connection is realized by passing the state of the previous layer to the subsequent layers. Let $F_{d,1}$ and $F_{d,5}$ be the input and output of the d-th CDB respectively. The output of d-th CDB can be formulated as

$$F_{d,5} = \sigma(w_{d,5}[F_{d,1}, F_{d,2}, F_{d,3}, F_{d,4}]) \tag{2}$$

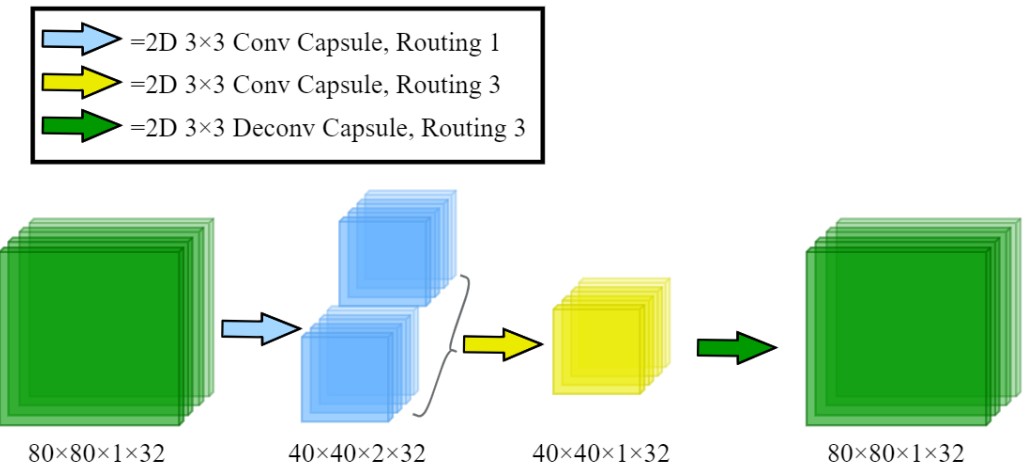

**Figure 2** **The architecture of our proposed Capsule Convolution Unit (CCU).** The input dimensions in the figure are length, width, number of capsules, and number of channels, respectively.

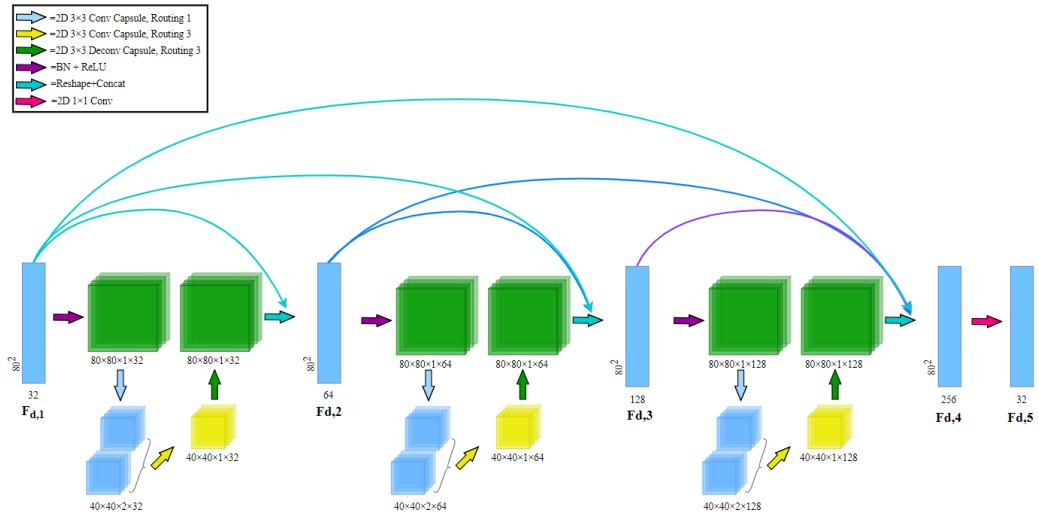

**Figure 3** **Capsule dense block (CDB) architecture.** Our CDB includes a dense capsule connection layer at the front and a regression layer at the rear. The dense capsule connection layer is composed of three Capsule Convolution Units (CCU) densely connected.

where $\sigma$ denotes the ReLU (*Glorot, Bordes & Bengio, 2011*) activation function. $w_{d,5}$ is the weights of the $F_{d,5}$, where the bias term is omitted for simplicity. $[\,F_{d,1}, F_{d,2}, F_{d,3}, F_{d,4}\,]$ refers to the concatenation of the feature maps produced by $F_{d,1}, F_{d,2}, F_{d,3}, F_{d,4}$. The CDB of the former layer and the output of each layer are directly connected with the latter layer, which not only retains the feedforward nature but also extracts the local dense feature.

## RESULTS

### DataSet

In our previous work, a total of three datasets have been used to evaluate our proposed NCDN model. They are the York University Cardiac MRI dataset, the Automated Cardiac Diagnosis Challenge, and the local dataset.

#### The York University Cardiac MRI dataset (York)

York consists of short-axis cardiac MR image sequences of 33 subjects, a total of 7980 2D images, provided by the Department of Diagnostic Imaging of the Hospital for Sick Children in Toronto, Canada (*Andreopoulos & Tsotsos, 2008*). Most data are a variety of cardiac abnormalities, such as cardiomyopathy, aortic regurgitation, ventricular enlargement, ischemia, *etc.*, and a few data are abnormalities related to the left ventricle. Because some of the markers in the dataset are missing or incomplete, the problematic images were removed from the dataset. The original $256 \times 256$ pixel images were clipped to form 3020 images with a scale of $80 \times 80$ pixels that only retained the left ventricle endocardium and epicardium.

#### The Automated Cardiac Diagnosis Challenge (ACDC)

The ACDC dataset was created based on real clinical examination results obtained by the University Hospital of Dijon (France) (*Bernard et al., 2018*). This dataset is the first and largest fully annotated public MRI cardiac data in the medical imaging community setting. The data consisted of short-axis section sequences of cardiac magnetic resonance images from 150 patients, divided into five subgroups, 30 normal subjects (NOR), 30 patients with previous myocardial infarction (MINF), 30 patients with dilated cardiomyopathy (DCM), 30 patients with hypertrophic cardiomyopathy (HCM), and 30 patients with abnormal right ventricle (RV). The spatial resolution was from 1.37 to 1.68 mm$^2$/pixel. We obtained 1902 images of 100 subjects from the training set of this dataset. Each slice was center cropped to a resolution of 128px by 128px.

#### Local dataset

This dataset has been employed in full left ventricle quantification (*Xue et al., 2017a*), and direct multitype cardiac indices estimation (*Xue et al., 2017b*). It consists of 2,900 images of 145 cases from three hospitals belonging to two medical centers (London Healthcare and St. Joseph's Healthcare). Most patients had a variety of pathological manifestations, including regional wall motion abnormalities, myocardial hypertrophy, mildly enlarged LV, atrial septal defect, LV dysfunction, *etc.*

### Metrics

Let $A$ and $M$ be the corresponding areas enclosed by the predicted (automated) contours $a$ and ground truth (manual) contours $m$, respectively. The following is our introduction to the main evaluation indicators.
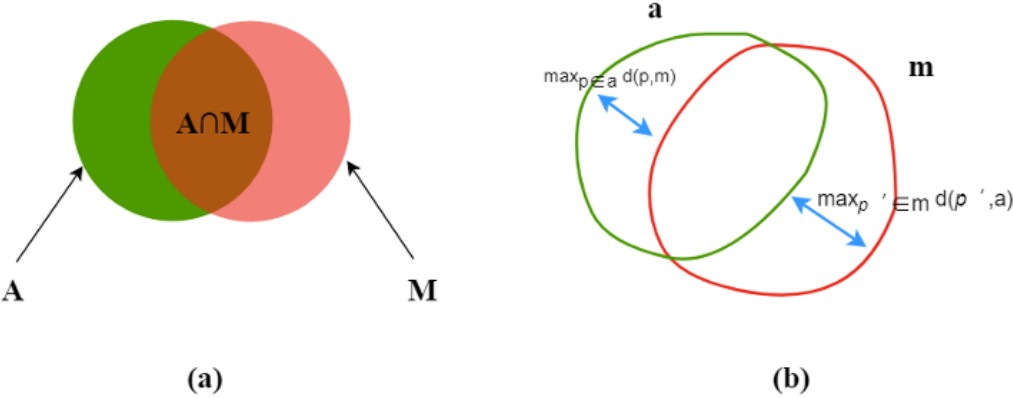

**Figure 4  Dice index (A) and Hausdorff distance (B).**

### Dice index

The Dice index (DI) (*Dice, 1945*) is a measure of overlap or similarity between two contour areas and is defined as (3) and Fig. 4A:

$$D(A, M) = 2 * \frac{A \cap M}{A + M}. \tag{3}$$

The Dice index varies from zero (total mismatch) to unity (perfect match).

### Hausdorff distance

Hausdorff distance (HD) (*Huttenlocher, Klanderman & Rucklidge, 1993*) is another evaluation metric, as shown in Fig. 4B. $d$ ($p,m$) means a point ($p$) on $a$ to its nearest point ($p'$) of m and the converse is $d$ ($p'$, $a$), as followed:

$$d(p, m) = \min||p - p'|| \tag{4}$$

then, find the maximum values of $d$ ($p,m$) and $d$ ($p'$, $a$) for all the points. *HD* is the maximum of the two values and always during [ 0, $\infty$]. *HD* increases, its performance degrades.

$$HD(m, a) = \max(\max d(p, m), \max d(p', a)). \tag{5}$$

## Training implementation

In the experiment, the deep learning framework is Tensorflow, the GPU is NVidia GTX 1080Ti, the optimizer is Adam, the loss function is cross-entropy loss, the learning rate is set to $10^{-4}$, the batch size is set to 1, and the training is about 30 epochs. Data expansion is applied to image expansion. For each dataset, we first divide the training set, validation set, and testing set. The division ratios are 0.7, 0.1, and 0.2. After that, the divided training set is expanded fourfold by rotating 90°, 180°, 270°.

**Table 2  To evaluate the effects of different segmentation techniques, the test results after training on the York data set are presented in the form of average (std.).**

| | York | | | |
|---|---|---|---|---|
| | Dice index | | HD (mm) | |
| | LV | MYO | LV | MYO |
| U-Net | 0.90 (0.03) | 0.95 (0.03) | 9.42 (3.7) | 8.32 (3.87) |
| FC-DenseNet | 0.91 (0.03) | 0.95 (0.03) | 9.15 (3.6) | 8.10 (3.83) |
| NCDN | **0.92 (0.03)** | **0.95 (0.02)** | **9.07 (3.49)** | **8.05 (3.62)** |

**Notes.**
Optimal values are indicated in bold.

**Table 3  To evaluate the generalization ability of different segmentation techniques, use the Local data set to test after training on the York dataset, and the results are presented in the form of average (std.).**

| | Local | | | |
|---|---|---|---|---|
| | Dice index | | HD (mm) | |
| | LV | MYO | LV | MYO |
| U-Net | 0.69 (0.09) | 0.85 (0.07) | 22.51 (6.58) | 21.55 (8.41) |
| FC-DenseNet | 0.74 (0.07) | 0.89 (0.06) | 18.93 (6.87) | **16.74 (5.88)** |
| NCDN | **0.78 (0.09)** | **0.89 (0.06)** | **14.94 (6.32)** | 17.13 (6.84) |

**Notes.**
Optimal values are indicated in bold.

## Generalization results

In this section, We show the results of the generalization ability of the segmentation model we designed. Table 2 is the test result of the three models trained on the York dataset. Table 3 is the result of training on the York dataset and testing on the Local dataset. Compared with the other two models, our model performed well in ordinary segmentation effect evaluation, which its Accuracy, Dice index, and Hausdorff distance all show the best results among the two contrasted models, as shown in Table 2. And as shown in Table 3, our model has better generalization ability than UNet, outperforms the FC-DenseNet on Segmentation of the LV and is on par with the FC-DenseNet on segmentation of the MYO. The model trained on the York dataset is used for testing on the Local dataset. Training with the York data set and then testing with the local data set results in worse segmentation than training and testing with the York data set. However, the NCDN we proposed still has better resistance to image changes and better performance than the other two models on DI, and HD.

## Ablation experiments

To verify the performance of the NCDN model, the following models were used for comparison: (1) The origin U-net model (*Ronneberger, Fischer & Brox, 2015*), (2) the FC-DenseNet introduced in *Jégou et al. (2016)*, (3) Nested Convolution Dense Network (NConvDN) replaces capsule convolution-capsule deconvolution of NCD-N with convolution-deconvolution. We used the ACDC dataset (*Bernard et al., 2018*) to train and test these models and separately count the segmentation effects of subjects with

**Table 4 Average DI and HD (std.) on RV in the four models.**

| | Dice index | | HD (mm) | |
| --- | --- | --- | --- | --- |
| | ED | ES | ED | ES |
| U-Net | 0.92 (0.09) | 0.86 (0.11) | 14.09 (11.63) | 20.66 (19.64) |
| FC-DenseNet | 0.92 (0.09) | 0.87 (0.12) | 14.40 (13.25) | 17.73 (17.14) |
| NConvDN | 0.92 (0.10) | 0.87 (0.11) | 14.80 (16.55) | 24.63 (34.73) |
| NCDN | **0.93 (0.07)** | **0.88 (0.10)** | **13.94 (12.23)** | **16.57 (14.95)** |

**Notes.**
Optimal values are indicated in bold.

**Table 5 Average DI and HD (std.) on MYO in the four models.**

| | Dice index | | HD (mm) | |
| --- | --- | --- | --- | --- |
| | ED | ES | ED | ES |
| U-Net | 0.87 (0.08) | 0.89 (0.07) | 11.90 (10.43) | 12.32 (12.08) |
| FC-DenseNet | 0.88 (0.06) | 0.89 (0.07) | **8.86 (4.03)** | **10.29 (7.11)** |
| NConvDN | 0.88 (0.07) | 0.89 (0.07) | 9.39 (6.85) | 11.71 (10.55) |
| NCDN | **0.89 (0.07)** | **0.90 (0.06)** | 8.91 (5.29) | 10.47 (6.27) |

**Notes.**
Optimal values are indicated in bold.

**Table 6 Average DI and HD (std.) on LV in the four models.**

| | Dice index | | HD (mm) | |
| --- | --- | --- | --- | --- |
| | ED | ES | ED | ES |
| U-Net | 0.95 (0.06) | 0.91 (0.09) | 9.33 (8.15) | 9.50 (7.37) |
| FC-DenseNet | 0.95 (0.05) | 0.91 (0.07) | 7.73 (3.56) | 9.16 (5.01) |
| NConvDN | 0.95 (0.08) | 0.91 (0.09) | 7.85 (4.89) | 9.29 (6.56) |
| NCDN | **0.96 (0.04)** | **0.92 (0.08)** | **7.71 (5.61)** | **8.64 (4.86)** |

**Notes.**
Optimal values are indicated in bold.

different disease types, which distinguished the end-diastolic (ED) and end-systolic (ES). The evaluation indicators include DI and HD. All four models follow the settings in 'Training Implementation' and use five-fold cross-validation. The results of comparing the NCDN model with other models according to the segmentation accuracy are shown in Tables 4, 5 and 6. The right ventricle is the hardest part of ventricular segmentation, yet through Table 4, NCDN outperforms the other three contrasting models in both Dice and HD metrics. Table 5 lists the segmentation results of myocardium. FC-DenseNet is slightly higher than NCDN in HD index, but NCDN is still the best performing model in Dice index. Table 6 shows the segmentation results of the left ventricle, compared with the other three comparison models, NCDN achieves the best results in both Dice and HD metrics.

Figure 5 shows the segmentation effect of the four models in the same image (*Bernard et al., 2018*). The images shown cover the ES and ED of different groups of people. From left to right are the original image, ground truth, and the prediction results of the four network models.

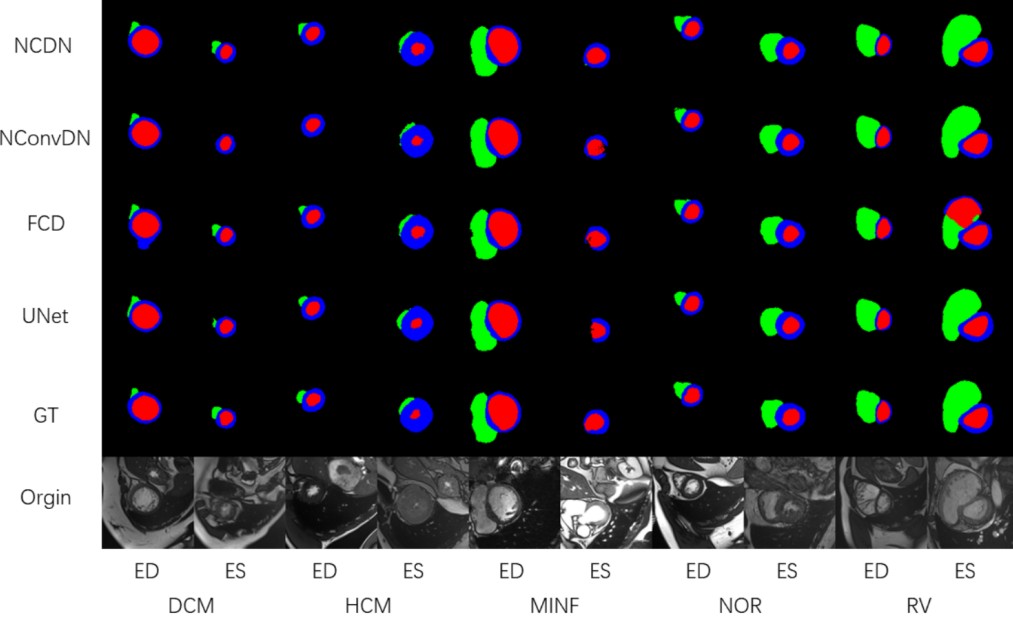

**Figure 5 The segmentation results of DCM, HCM, MINF, NOR, and RV.** The illustration shows the segmentation effect of NCDN and the other three networks, where GT stands for ground truth (*Bernard et al., 2018*). The data shown in the figure comes from the ACDC dataset (*Bernard et al., 2018*).

Using UNet, FC-DenseNet, NConvDN, and the proposed NCDN to conduct ablation experiments, the results show that our proposed model has better performance. In the comparison of multiple dimensions, the segmentation accuracy of NConvDN is worse than that of NCDN. This may be because capsule convolution can save more spatial information than convolution, which is beneficial to improve the ability to distinguish things from different perspectives.

In order to better prove the performance of the proposed model, the model with the most advanced results was used for comparison, which ranks among the top 10 in ACDC test set segmentation performance. As shown in Table 7(A), on the Dice metric, the NCDN obtained the best results on ED of MYO and ED of LV compared with the other ten models. In addition, ED and ES of RV ranked 6th, ES of MYO ranked 2nd, and ES of LV ranked 4th. As shown in Table 7(B), in terms of HD index, ED and ES of RV ranked 8th and 7th respectively, ED and ES of MYO ranked 6th, and ED and ES of LV ranked 6th and 7th respectively. NCDN does not perform very well in HD, using some proper post-processing methods may bring improvements.

## DISCUSSION

In this work, a nested network structure is proposed to complete the segmentation task of cardiac MRI, hoping to have better segmentation performance. Local and third-party evaluations have reflected that it has improved segmentation accuracy and robustness relative to the benchmark model, and it also has the most advanced results in some indicators. The proposed model has good performance on the DI index, but the

**Table 7  The segmentation effect of different segmentation techniques on the ACDC test set.**

| | Dice index | | | | | |
| | RV | | MYO | | LV | |
| | ED | ES | ED | ES | ED | ES |
|---|---|---|---|---|---|---|
| NCDN | 0.932 | 0.882 | **0.899** | 0.911 | **0.966** | 0.916 |
| *Isensee et al. (2018)* | **0.946** | **0.904** | 0.896 | **0.919** | 0.965 | **0.933** |
| *Zotti et al. (2018)* | 0.934 | 0.885 | 0.886 | 0.902 | 0.964 | 0.912 |
| *Painchaud et al. (2020)* | 0.933 | 0.884 | 0.881 | 0.897 | 0.961 | 0.911 |
| *Khened, Alex & Krishnamurthi (2018)* | 0.941 | 0.882 | 0.889 | 0.898 | 0.964 | 0.917 |
| *Baumgartner et al. (2017)* | 0.935 | 0.879 | 0.882 | 0.897 | 0.963 | 0.911 |
| *Wolterink et al. (2017)* | 0.932 | 0.883 | 0.892 | 0.901 | 0.961 | 0.918 |
| *Rohé, Sermesant & Pennec (2017)* | 0.928 | 0.872 | 0.884 | 0.896 | 0.957 | 0.900 |
| *Zotti et al. (2017)* | 0.916 | 0.845 | 0.875 | 0.894 | 0.957 | 0.905 |
| *Patravali, Jain & Chilamkurthy (2017)* | 0.911 | 0.819 | 0.867 | 0.869 | 0.955 | 0.885 |
| *Grinias & Tziritas (2018)* | 0.887 | 0.767 | 0.799 | 0.784 | 0.948 | 0.848 |

| | HD (mm) | | | | | |
| | RV | | MYO | | LV | |
| | ED | ES | ED | ES | ED | ES |
|---|---|---|---|---|---|---|
| NCDN | 13.9 | 14.5 | 9.7 | 10.8 | 7.5 | 9.6 |
| *Isensee et al. (2018)* | **8.8** | **11.4** | **7.6** | **7.1** | **5.6** | **6.3** |
| *Zotti et al. (2018)* | 11.0 | 12.6 | 9.6 | 9.3 | 6.2 | 8.4 |
| *Painchaud et al. (2020)* | 13.7 | 13.3 | 8.6 | 9.6 | 6.1 | 8.3 |
| *Khened, Alex & Krishnamurthi (2018)* | 10.3 | 14.0 | 9.8 | 12.6 | 8.1 | 9.0 |
| *Baumgartner et al. (2017)* | 14.0 | 13.9 | 9.8 | 11.3 | 6.5 | 9.2 |
| *Wolterink et al. (2017)* | 12.7 | 14.7 | 8.7 | 10.6 | 7.5 | 9.6 |
| *Rohé, Sermesant & Pennec (2017)* | 11.9 | 13.4 | 8.7 | 9.3 | 7.5 | 10.7 |
| *Zotti et al. (2017)* | 14.0 | 15.9 | 11.1 | 10.7 | 6.6 | 8.7 |
| *Patravali, Jain & Chilamkurthy (2017)* | 13.5 | 18.7 | 11.5 | 13.0 | 8.2 | 10.9 |
| *Grinias & Tziritas (2018)* | 19.0 | 24.2 | 12.3 | 14.6 | 8.9 | 12.9 |

**Notes.**
Optimal values are indicated in bold.

performance is relatively ordinary on the HD index, especially for the segmentation of RV. Both the DI indicator and HD indicator measure the effect of segmentation but have different focuses. DI can better reflect the consistency of the corresponding pixels of the image, while HD focuses on the consistency of the segmentation edge. This means that an outlier has little effect on DI, but may have a greater impact on HD indicators. NCDN is an end-to-end network with capsule convolution as the kernel. The characteristics of the capsule enable it to better perceive objects in different viewing angles, which effectively avoids the state of failure of recognition in FC-DenseNet, such as incorrectly judging whether a certain category exists on the image. However, it still lacks constraints on the objects to be segmented in the image, which may lead to the appearance of outliers and make the HD index too large. To alleviate this problem, new shape mechanisms such as shape prior (*Ravishankar et al., 2017*) can be introduced. In addition, a more reasonable

cost function for HD constraints can be discovered or statistical techniques can be used to correct outliers in the segmented image to ensure edge integrity and internal consistency.

## CONCLUSIONS

In this work, we propose a nested network structure that decomposes a single codec into multiple codecs to obtain better cardiac MR image segmentation results. This structure based on the FC-DenseNet model and capsule convolution-capsule deconvolution shows a better segmentation effect on multiple datasets than each part of the source network. The smaller error and standard deviation further prove the effectiveness of network fusion. The experimental results show that the segmentation effect of our model has better stability than other traditional segmentation models. This makes it possible to apply our method to the automatic segmentation of cardiac MRI systems in the future.

### Funding

This work was supported by the National Science Foundation Program of China (NSFC) under Grants 61976241, 61871173, by major research program of National Science Foundation of China (NSFC) under Grant 91948303, by the Tianjin Science and Technology Planning Project under Grants 19ZXJRGX00080, by Science and Technology Program of Tianjin under Grant 20YDTPJC00670, and by the project of the Left and right ventricle segmentation method of cardiac MRI images, Shanxi Key Laboratory of Biomedical Imaging and Image Big Data, North University of China. There was no additional external funding received for this study. The funders had no role in study design, data collection and analysis, decision to publish, or preparation of the manuscript.

### Grant Disclosures

The following grant information was disclosed by the authors:
National Science Foundation Program of China (NSFC): 61976241, 61871173.
National Science Foundation of China (NSFC): 91948303.
Tianjin Science and Technology Planning Project: 19ZXJRGX00080.
Science and Technology Program of Tianjin: 20YDTPJC00670.
Left and right ventricle segmentation method of cardiac MRI images, Shanxi Key Laboratory of Biomedical Imaging and Image Big Data, North University of China.

### Competing Interests

The authors declare there are no competing interests.

### Author Contributions

- Jilong Zhang conceived and designed the experiments, performed the experiments, analyzed the data, performed the computation work, authored or reviewed drafts of the article, and approved the final draft.

- Yajuan Zhang conceived and designed the experiments, performed the experiments, analyzed the data, authored or reviewed drafts of the article, and approved the final draft.
- Hongyang Zhang performed the experiments, analyzed the data, performed the computation work, prepared figures and/or tables, and approved the final draft.
- Quan Zhang conceived and designed the experiments, analyzed the data, authored or reviewed drafts of the article, and approved the final draft.
- Weihua Su performed the experiments, performed the computation work, prepared figures and/or tables, and approved the final draft.
- Shijie Guo performed the experiments, performed the computation work, prepared figures and/or tables, and approved the final draft.
- Yuanquan Wang conceived and designed the experiments, analyzed the data, authored or reviewed drafts of the article, and approved the final draft.

## Data Availability

The source code is available at GitHub: https://github.com/jk1008611/NCDN.

The data is available from Automated Cardiac Diagnosis Challenge (ACDC): https://acdc.creatis.insa-lyon.fr/description/results.html.

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
