# Peer review of "Segmentation of biventricle in cardiac cine MRI via nested capsule dense network"

_PeerJ Computer Science, doi:10.7717/peerj-cs.1146_

## Round 0.1 · original submission · Major Revisions

Three detailed reports have been received. The reviewers are generally positive but they also raised some issues that need to be addressed in the revision. Please provide a detailed response letter. Thanks.

Reviewer 1 ·

Basic reporting

The authors evaluate a nested capsule dense architecture for bi-ventricular segmentation on cardiac MRI cines.

It is suggested to introduce more existing work about the segmentation of LV and RV based on the deep learning in the introduction/related work part

Line 66: "there are still many problems to be solved". Please specify what these problems are and how they could be solved.

Experimental design

no comment

Validity of the findings

no comment

Additional comments

The FCN and U-net have been proposed for several years, suggest the author contrasts the proposed method with some state-of-the-art segmentation networks, rather than just the U-net or DenseNet

The authors are asked to provide more details of how the database have been constructed (Were trabeculae or papillary muscles included? Were all phases segmented or just ED/ES?...). It is quite important that the reader understands the quality of the data.

To provide an unbiased evaluation of a model, are the datasets divided into training, validation and test sets on a patient-level ?

"...as shown in Table 3, our model has better 278 generalization ability." Is this statement valid considering FC-DenseNet leads to slightly better result for Myo in term of HD than NCDN, while DenseNet and NCDC achived the same accuracy with an average Dice index of 0.89 for LV ?

Line 266 is repetitive.

Are results reported in table 2 and 3 based on five-fold cross validation?

Some results have no normal distribution. Therefore they should be reported as the median (interquartile range) rather than average (SD)

Code is not available and so the work is not reproducible. I suggest making the code available on GitHub (with a DOI) - or add a conflict of interest statement if the Institution/authors plan to commercialise it.

Reviewer 2 ·

Basic reporting

The manuscript needs proofreading to deliver a clear explanation for readers.

Experimental design

no comment

Validity of the findings

no comment

Additional comments

The authors have introduced well contribution, however, the paper needs some improvement. The review comments are listed below.
1. In line 40 you started the sentence in an improper way by [2,3]. You must start the sentence clearly. For example, name the proposed methods or authors’ names and then references numbers.
2. In line 43 you may modify the sentence properly. For example, Martí et al. [4] proposed research on differential privacy to avoid the privacy problems......
3. In Line 47 you should list the regression methods with their corresponding references and restructure the sentence in a proper way.
4. As you followed IEEE formatting for references, please be sure of including the year after the authors’ names as in lines 124, 126, 135, 148, 156, and 192.
5. Reconstruct the sentence in line 240 by clarifying the method or authors, not only mentioning the references directly.
6. Clarify why you used batch size 1 rather than 4,8, and so on?
7. Instead of ‘in section 4.4’ write in this section (line 271).
8. Tables 4,5 and 6 need more discussion, and table 7 needs proper discussion.
9. References section deviates from the journal template if it is compulsory to follow. As in the guidelines, you can use any references formatting, but recheck the author's name properly in the reference section.

Reviewer 3 ·

Basic reporting

The paper addresses the segmentation of ventricles in cardiac cine-MRI using a proposed network called Nested Capsule Dense Network (NCDN). The proposed method was tested on three datasets: York Cardiac Dataset, ACDC, and a local dataset. In both tests, the results achieved were promising, indicating the feasibility of applying the method in the scenario of cardiac segmentation. The authors also performed a comparative analysis between the results of NCDN with other networks, including U-Net, which is one of the main methods used in segmentation of medical images. In the case of the ACDC dataset, the authors presented the results obtained in the online evaluation platform of the challenge, in which the proposed method appears with a good classification.

Experimental design

The following points should be considered:

1 – Line 94: The mentioned contribution is unclear. How does the connection structure work in the model on which the proposed network is based? What is the impact of the proposed change?

2 - The organization of the Related Work (Section 2) could be improved. The authors present the works divided into very broad categories (Convolutional Neural Networks and Capsule Neural Networks). It would be interesting to present the works related to cardiac segmentation in another subsection, explaining the techniques used, and emphasizing the main differences in relation to the proposed method.

3 – Regarding the Datasets (Section 4.1), it is interesting to add figures that demonstrate the cardiac images and their respective annotations (ground-truth).

4 - Is the centralized crop of the slices a step of the proposed method? As this process is based only on the center of the image, is not there a risk that the cardiac structures will be partially cropped?

5 – Regarding the data expansion, which rotation angles were used? Why did the authors not consider using other operations?

6 - Figure 5 could indicate more clearly which are the segmented cardiac structures. One suggestion is to add a colored legend to the figure to indicate Myo, RV, and LVC. In addition, it is important that authors also include figures that present qualitative results for the other datasets (York and Local).

Validity of the findings

- The authors should clarify the strengths and weaknesses of the proposed method compared to the related works.

- The ACDC challenge assesses geometric (Dice coefficient and Hausdoff distance) and clinical metrics on its online platform. It is important that authors present all metrics.

Additional comments

An English review is recommended to track errors and colloquial expressions, as written on line 58.

---

## Round 0.2 · Major Revisions

There is still some work to be done. A further revision is needed.

Reviewer 1 ·

Basic reporting

no comment

Experimental design

no comment

Validity of the findings

no comment

Additional comments

Zhang and colleagues underwent a significant effort to meet the reviewers1 comments. However, some
items still need to be addressed:

1- As I previously mentioned on my comment number 1, the introduction/related work section should be better structured. It is suggested to introduce more existing work about the segmentation of LV and RV based on the deep learning. Authors should better specify the limits of the current literature and how they intend to overcome these limitations. Specifically, how are researchers addressing the problem of improving the generalization ability of the model in the context of limited datasets? While it is clear how the authors are dealing with the problem of improving the model generalization ability, it is not clear how their method differs from that of existing works.

2- In response to my previous comment number 6, authors said NCDN model is "significantly" better than fc-denseNet model. To evaluate a significant difference between models, authors should introduce a statistical test. Which differences are statistically significant by p-values?

3- Please provide the link to the corresponding Github page in the text.

Reviewer 3 ·

Basic reporting

The authors have answered most of the questions satisfactorily.
Some minor reviews are requested:

1 - It is recommended to report in Section 4.3 which rotation angles were used in the data expansion process.

2 - it is important that authors also include figures that present the segmentation results for the other datasets (York and Local).

Experimental design

no comment

Validity of the findings

no comment

Additional comments

no comment

---

## Round 0.3 · accepted · Accept

The authors have addressed all reviewers' comments so the paper can be accepted.

Reviewer 1 ·

Basic reporting

no comment

Experimental design

no comment

Validity of the findings

no comment

Additional comments

The Authors have exhaustively replied to all my questions. I have no further comments.

Reviewer 2 ·

Basic reporting

The authors have done previous comments correctly.
But they need to clarify why they used a mini-batch size of 1 rather than 4, 8, 16 and etc.
Authors also need to check body text alignment in some paragraphs.

Experimental design

The experimental design was designed clearly.

Validity of the findings

The outcomes of this paper are valid with good performance.

Reviewer 3 ·

Basic reporting

The authors have answered all questions satisfactorily.

Experimental design

No more questions

Validity of the findings

No more questions

Additional comments

No more questions